# Metabolome-based prediction of yield heterosis contributes to the breeding of elite rice

Zhiwu Dan , Yunping Chen, Weibo Zhao, Qiong Wang, Wenchao Huang

Improvement of the breeding efficiencies of heterotic crops adaptive to different conditions can mitigate the food shortage crisis due to overpopulation and climate change. To date, diverse molecular markers have been used to guide field phenotypic selection, whereas accurate predictions of complex heterotic traits are rarely reported. Here, we present a practical metabolome-based strategy for predicting yield heterosis in rice. The dissection of population structure based on untargeted metabolite profiles as the initial critical step in multivariate modeling performed better than the screening of predictive variables. Then the assessment of each predictive variable's contribution to predictive models according to all latent factors was more precise than the conventional first one. Metabolites belonging to specific pathways were closely associated with yield heterosis, and the up-regulation of galactose metabolism promoted robust yield heterosis in hybrids under different growth conditions. Our study demonstrates that metabolome-based predictive models with correctly dissected population structure and screened predictive variables can facilitate accurate predictions of yield heterosis and have great potential for establishing molecular marker–based precision breeding programs.

## Introduction

Hybrid rice has a higher yield than conventional rice and has contributed greatly to boosting global food production over the past four decades (Cheng et al, 2007). However, to feed 9.1 billion people by 2050 (FAO, 2009), the rate at which the annual yield of staple crops increases must be more than double (Hickey et al, 2017). Worse yet, severe global yield losses of rice, maize, and soybean are plausible under dramatic climate change (Zhao et al, 2016, 2017). Therefore, the breeding of new heterotic varieties adaptive to different growth conditions is a promising option for ensuring global food security.

Elite hybrid rice, such as the wild abortive and Honglian series (Luo et al, 2013; Huang et al, 2015), are perfect combinations of sterile and restorer lines for generating yield heterosis through successful breeding (Chen et al, 2014). The yield potential of a new combination is unknown before field hybridization and phenotyping, which are labor-intensive and time-consuming tasks. To improve breeding efficiency, biomarkers, including DNA, RNA, proteins, and metabolites, that can reflect parental genetic differences have been adopted for crop improvement. The results of marker-based performance prediction in maize, rice, wheat, sorghum, and potato demonstrate that various factors, including genetic relatedness (Zhao et al, 2015; Yu et al, 2016), environmental differences (Li et al, 2018; Sprenger et al, 2018), predictive variable number (Technow et al, 2014; de Abreu et al, 2017; Sprenger et al, 2018), feature selection method (Xu et al, 2014; de Abreu et al, 2017), and population structure and size (Riedelsheimer et al, 2012a; Xu et al, 2014; Zhao et al, 2015), can affect the power of predictive models. In addition, the absence of high-quality reference genomes and the genome complexity of polyploids also hinder the establishment of highly efficient molecular breeding programs (Rasheed et al, 2017).

The metabolome is a bridge between the genome and phenome, and metabolite levels have close and comprehensive connections with polygenic complex traits (Schauer et al, 2006; Riedelsheimer et al, 2012a, 2012b). Metabolic prediction of agronomic traits has been preliminarily explored in maize and rice (Dan et al, 2016; Xu et al, 2016; de Abreu et al, 2017), and metabolic analytes detected in young rice seedlings can predict grain weight in hybrids from different populations (Dan et al, 2019). However, the accurate prediction of yield heterosis remains a challenge because the mechanisms of yield heterosis are highly complex and the corresponding component traits (grain weight, grain number per panicle, tiller number, and seed setting rate) are polygenic. Moreover, the performance of complex quantitative traits such as yield and disease resistance is determined by the genotype, environment, and their interaction (Thomas, 2010; Scheres et al, 2017; Li et al, 2018). Here, we provide a practical metabolome-based strategy for predicting yield heterosis in rice that has wide applicability and can contribute to the breeding of climate-resilient hybrid crops,

---

State Key Laboratory of Hybrid Rice, Key Laboratory for Research and Utilization of Heterosis in Indica Rice, The Yangtze River Valley Hybrid Rice Collaboration & Innovation Center, College of Life Sciences, Wuhan University, Wuhan, China

Correspondence: wenchaoh@whu.edu.cn

especially those of orphan varieties without reference genomes or polyploidy.

# Results

### Determination of predictive variables for yield heterosis

To build predictive models for rice hybrids under different growth conditions, we measured the grain yield of two $F_1$ hybrid populations (Pop2012 and Pop2015) and calculated better-parent heterosis for yield per plant (BPH-YPP) for subsequent analysis (Table S1). The YPPs of parents from the two populations were significantly different (Fig 1A and Table S2), and hybrid YPP and BPH-YPP also differed significantly between populations (Fig 1B and Table S2). Then, we performed untargeted liquid chromatography–mass spectrometry (LC-MS) analysis of 15-d-old seedlings of the parents of both populations, and a total of 3,746 metabolic analytes were detected in the parents (Table S3). A dendrogram of parents from Pop2012 based on parental metabolite profiles showed a clustering trend similar to that based on *indica–japonica*–specific insertion and deletion DNA markers (Figs 1C and S1) (Dan et al, 2016). Moreover, principal component analysis (PCA) (Jolliffe, 1986) of the metabolome of the parents from both populations revealed little overlaps between populations (Fig 1D).

To determine predictive variables for yield heterosis, we then performed metabolite profiling of seedlings of three pairs of reciprocal $F_1$ hybrids and investigated the relationship between hybrid metabolite profiles and corresponding transformed parental metabolite levels (Table S3). The means of, differences in, and ratios of parental metabolite levels were calculated for each hybrid, and the ratios of parental metabolite levels were highly distinct from the hybrid metabolite profiles according to PCA score plots (Fig 1E and Table S4). In contrast, the means of and differences in parental metabolite levels showed closer connections, and the parental means displayed the distributions most similar to the hybrid metabolite profiles (Fig 1F and Table S4), suggesting that mean parental metabolite levels are appropriate for representing hybrid metabolite profiles.

Subsequently, the means of and differences in all detected analytes were calculated between every pair of parents for the two populations and assigned as predictive variables to corresponding hybrids. Partial least squares (PLS) regression (Wold, 1982), which can model quantitative and multivariate complex relationships, was applied to the predictive variables and BPH-YPP for hybrids from Pop2012. No more than 17 latent factors were extracted, and the adjusted R-square values of the means used as predictive variables were much larger than those of the differences (Fig 1G). Furthermore, PCA of the mean metabolite levels of the parents of hybrids demonstrated distant genetic relationships between populations (Fig 1H), suggesting that the materials were suitable for reliable cross-validation (Wray et al, 2013). In summary, these results show that the means of parental metabolite levels are suitable predictive variables for yield heterosis and the two populations used in this study are suitable for exploring predictive models for genetically distant hybrids under different growth conditions.

### Prediction of BPH for grain yield across environments

To predict BPH-YPP for Pop2015 based on Pop2012, parameters of predictive analytes and the constant in Fig 1G (three latent factors with an adjusted R-square value of 0.3647) were used to construct an equation for predicting yield heterosis. The predictability of BPH-YPP was 0.61 for Pop2012 (Fig 2A; $P = 5.7 \times 10^{-31}$) but only 0.24 for Pop2015 (Fig 2B; $P = 0.01$). Then, quantiles of BPH-YPP for Pop2012 were calculated to divide the hybrids into high- and low-BPH-YPP subgroups to increase predictability by grouping appropriate predictive variables (Fig 2C). The metabolome-based PLS-discriminant analysis (PLS-DA) (Barker et al, 2003) grouped hybrids with high- and low BPH-YPP separately (Fig S2). Next, the top 1,000 predictive variables were chosen for modeling based on the variable importance in projection (VIP) of component 1 (VIP > 1.0198; Table S5). Unexpectedly, although the predictability of BPH-YPP for Pop2012 was as high as 0.61 (Fig 2D), the predictability of BPH-YPP for Pop2015 was only 0.17 (Fig 2E; $P = 0.09$). We obtained even lower predictabilities of BPH-YPP for Pop2015 by changing the number of predictive variables (Fig 2F and G; VIP > 1.4631, 500 predictive analytes) and the numbers of high- and low-BPH-YPP hybrids from Pop2012 for screening predictive variables (Figs S3–S5).

Next, PCA was performed on both populations based on the top 1,000 predictive variables selected as shown in Fig 2D (Fig S6), and the principal component 1 scores were ordered to identify core hybrids (Tables S6 and S7). Finally, one of every three hybrids from Pop2012 (herein named the 1/3N set) and one of every two hybrids from Pop2015 (1/2N set) were combined as the training set (1/3N + 1/2N set) to predict the remaining hybrids. However, the predictability for Pop2015 was only 0.16 (Fig 3A; $P = 0.26$). Clues from the slight predictability changes in Pop2012, as shown in Fig 2D ($r = 0.61$) and Fig 3A ($r = 0.51$), indicated that the low predictability of BPH-YPP for Pop2015 might have been caused by improper selection of core hybrids.

The proportions of variance explained by the first two principal components were smaller than those shown in Fig S6A and C (Fig S6B and D and Tables S8 and S9), indicating that the PCA of all 3,746 predictive variables rather than the top 1,000 predictive variables helped dissect population structure. The predictability of BPH-YPP for Pop2015 increased to 0.34 when the top 1,000 predictive variables were used (Fig 3B; $P = 0.01$). Next, the performance of "noncore" hybrids from both populations was predicted with all 3,746 predictive variables. A correlation analysis between the observed and predicted BPH-YPPs indicated that the predictability of BPH-YPP for Pop2015 was further improved (Fig 3C; $r = 0.44$, $P = 0.001$). Therefore, for predicting component traits (e.g., grain weight) (Dan et al, 2019), the dissection of population structures as the initial step in building predictive models is better than the screening of predictive variables.

### Optimization and validation of the predictive model

To further increase the predictability of BPH-YPP, we filtered low-contribution or unrelated predictive variables among the 3,746 analytes based on VIP values of the first latent factor (Table S10). As shown in Fig S7A, the predictabilities of BPH-YPP for Pop2015 varied with changes in the number of predictive variables. Then, the mean

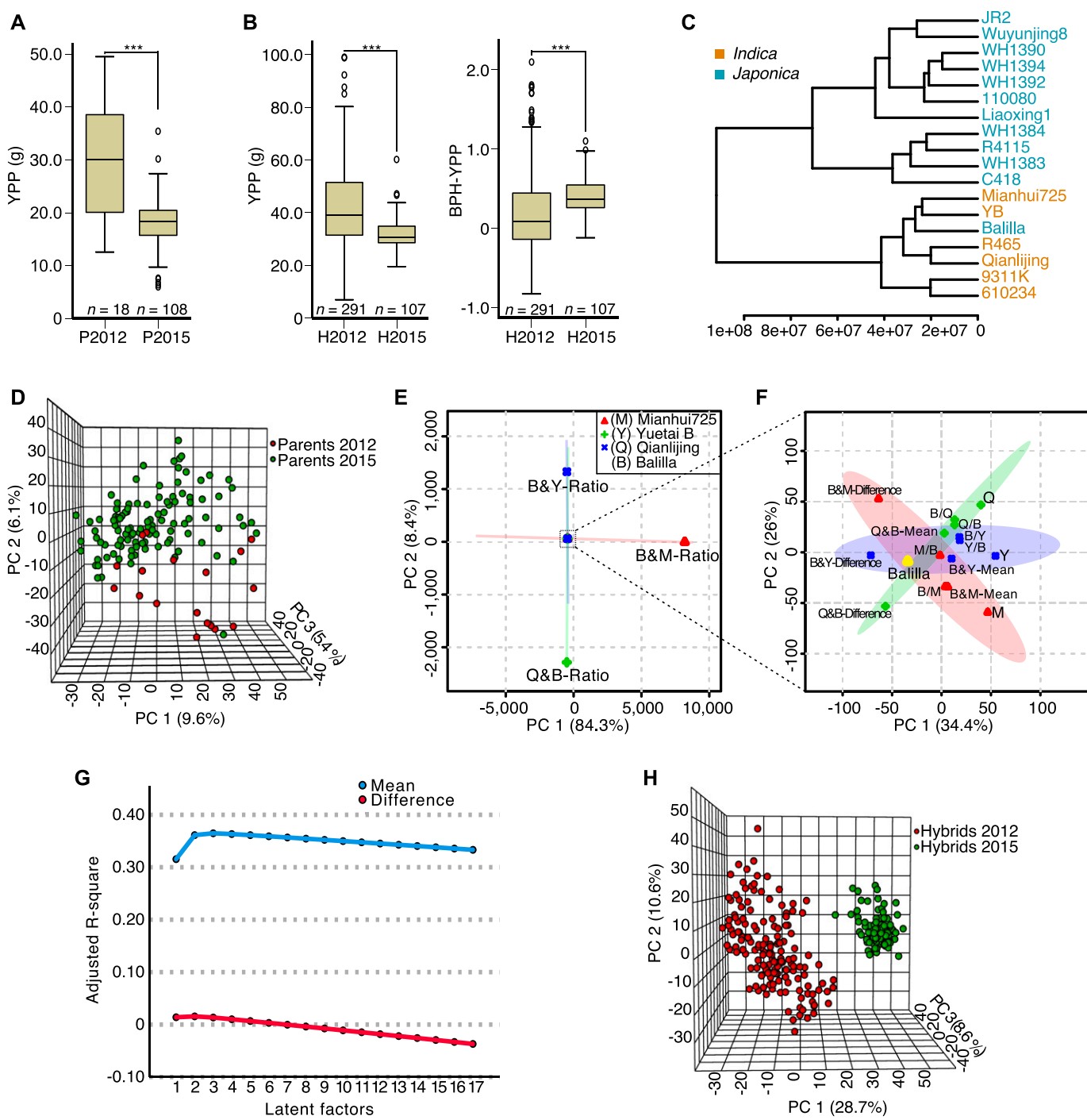

**Figure 1. Determination of predictive variables for BPH-YPP.**

**(A)** YPP of parents from Pop2012 and Pop2015. ***$P$ = 2.7 × $10^{-4}$, two-tailed, independent-samples $t$ test. **(B)** YPP and BPH-YPP of hybrids from Pop2012 and Pop2015. For YPP, ***$P$ = 2.1 × $10^{-19}$, two-tailed, independent-samples $t$ test. For BPH-YPP, ***$P$ = 5.2 × $10^{-7}$, two-tailed, independent-samples $t$ test. **(C)** Dendrogram of the 18 cultivars from Pop2012 based on the 3,746 detected metabolic analytes. **(D)** Three-dimensional distribution of parents from Pop2012 and Pop2015 based on PCA of the 3,746 detected analytes. **(E, F)** Distributions of three pairs of reciprocal hybrids and corresponding parents based on PCA of 3,746 metabolic analytes. The ratios of (E), means of, and differences in (F) parental metabolite levels were calculated and compared to hybrid metabolite profiles. Slashes indicate crosses between two parents, and female and male parents are on the left and right of the slash, respectively. Shadows with color are 95% confidence regions. **(G)** Changes in the number of latent factors and adjusted R-square values. PLS regression analysis was performed between the BPH-YPP of hybrids from Pop2012 and the means of/differences in parental metabolite profiles. No more than 17 latent factors were extracted in both analyses. **(H)** Three-dimensional distribution of hybrids from Pop2012 and Pop2015 based on PCA of the 3,746 predictive variables.

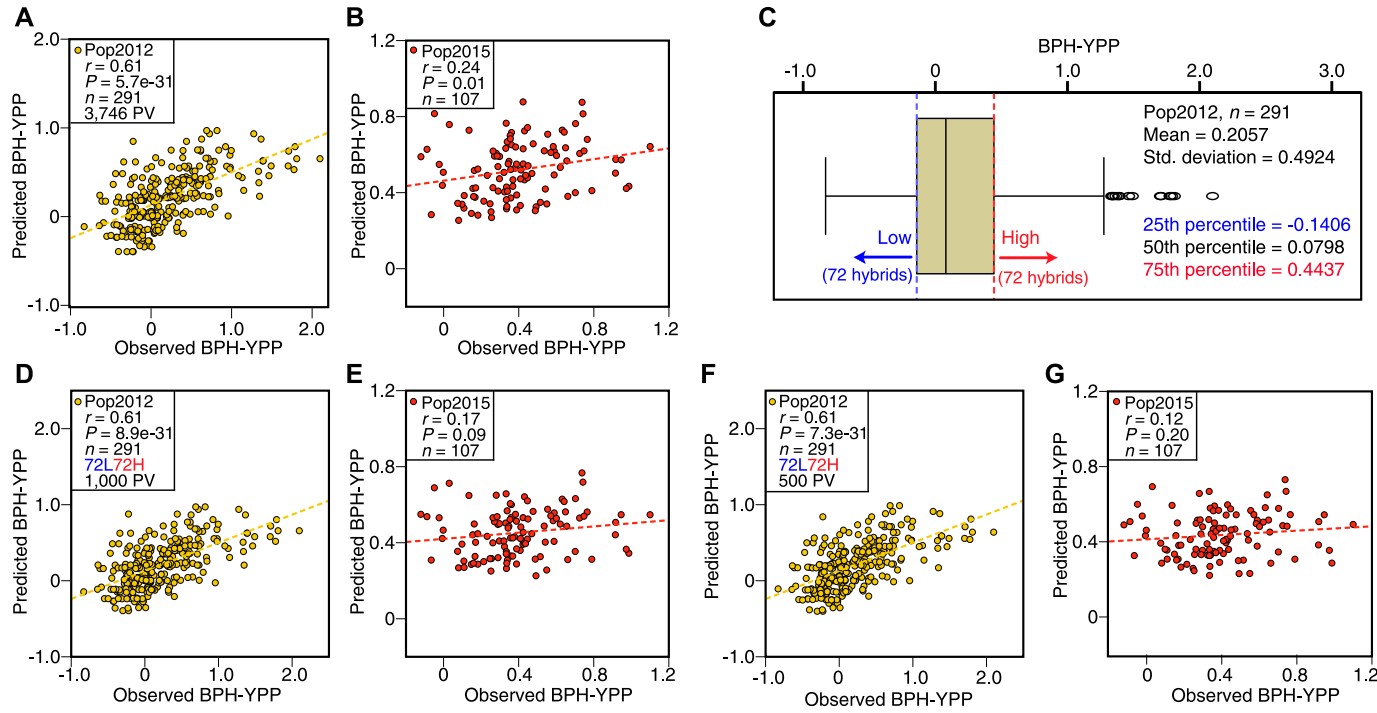

**Figure 2.  Prediction of BPH-YPP with predictive variables selected from PLS-DA.**
**(A)** Prediction of BPH-YPP for Pop2012 with 3,746 predictive variables (PVs). **(B)** Prediction of BPH-YPP for Pop2015 based on the predictive model constructed in Fig 2A. **(C)** Box plot of BPH-YPP for Pop2012. Hybrids were divided into low- and high-BPH-YPP subgroups according to the 25th and 75th percentiles. **(D)** Prediction of BPH-YPP for Pop2012 with the top 1,000 predictive variables based on PLS-DA of 72 high- and 72 low-BPH-YPP hybrids. **(E)** Prediction of BPH-YPP for Pop2015 based on the predictive model constructed in Fig 2D. **(F)** Prediction of BPH-YPP for Pop2012 with the top 500 predictive variables based on PLS-DA of 72 high- and 72 low-BPH-YPP hybrids. **(G)** Prediction of BPH-YPP for Pop2015 based on the predictive model constructed in Fig 2F. The dotted lines are the fit lines.

VIP values of the first three latent factors were calculated (Table S10), and the predictabilities did not increase (Fig S7B). Unexpectedly, the predictabilities of BPH-YPP for Pop2015 increased when all seven latent factors were used (Fig 4A), peaking at 0.58 with 1,400 predictive variables ($P = 5.0 \times 10^{-6}$; Fig 4B and Table S11). Thus, hybrids from both populations were predicted with predictabilities close to 0.6, although the ranges of yield heterosis differed greatly between the two populations. Moreover, no improvement in predictability was observed after another 400 predictive variables were removed from the 1,400 correlated predictive analytes (Fig S8A and B; $r = 0.55$, $P = 1.9 \times 10^{-5}$).

We next performed $t$ tests for high- and low-BPH-YPP hybrids from Pop2012 according to the results shown in Fig 2C. A total of 1,311 analytes were significantly different between the two subgroups

(Fig 4C and Table S12), and most of them overlapped with the 1,400 predictive variables (Fig 4D). The results from MetaboAnalyst (Xia et al, 2011) indicated that metabolic pathways, including that for galactose metabolism, were enriched for both populations (Fig 4E and F, Tables S13 and S14, Figs S9, and S10), which was consistent with findings in previous reports (Schauer et al, 2006; Obata et al, 2015; Wen et al, 2015). In addition, galactose metabolism was detected when dysregulated metabolic pathway analysis was performed on high- and low-BPH-YPP hybrids from Pop2012 with MetDNA (Shen et al, 2019) (Fig 4G and Table S15). The average levels of metabolites involved in galactose metabolism in high-BPH-YPP hybrids were higher than those in low-BPH-YPP hybrids (Fig 4H, Tables S16, and S17), and significant differences in metabolites involved in galactose metabolism were detected between the two subgroups (Table S18).

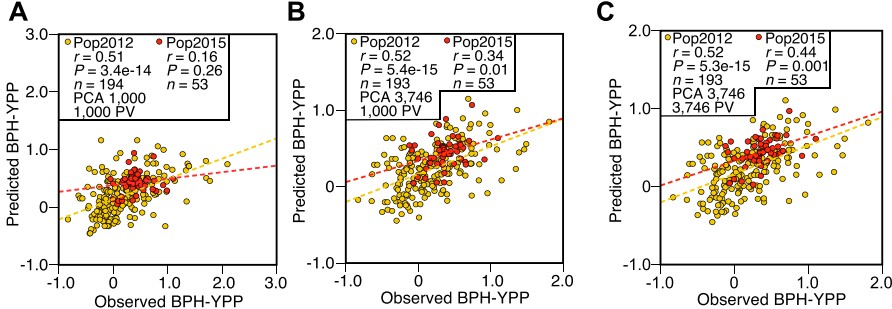

**Figure 3.  Dissection of population structure increases predictabilities.**
**(A)** Prediction of BPH-YPP for Pop2012 and Pop2015 with the top 1,000 predictive variables used in Fig 2D and population structure determined by the top 1,000 predictive variables. **(B)** Prediction of BPH-YPP for Pop2012 and Pop2015 with the top 1,000 predictive variables used in Fig 2D and population structure determined by 3,746 predictive variables. **(C)** Prediction of BPH-YPP for Pop2012 and Pop2015 with 3,746 predictive variables and population structure determined by 3,746 predictive variables. The dotted lines are the fit lines.

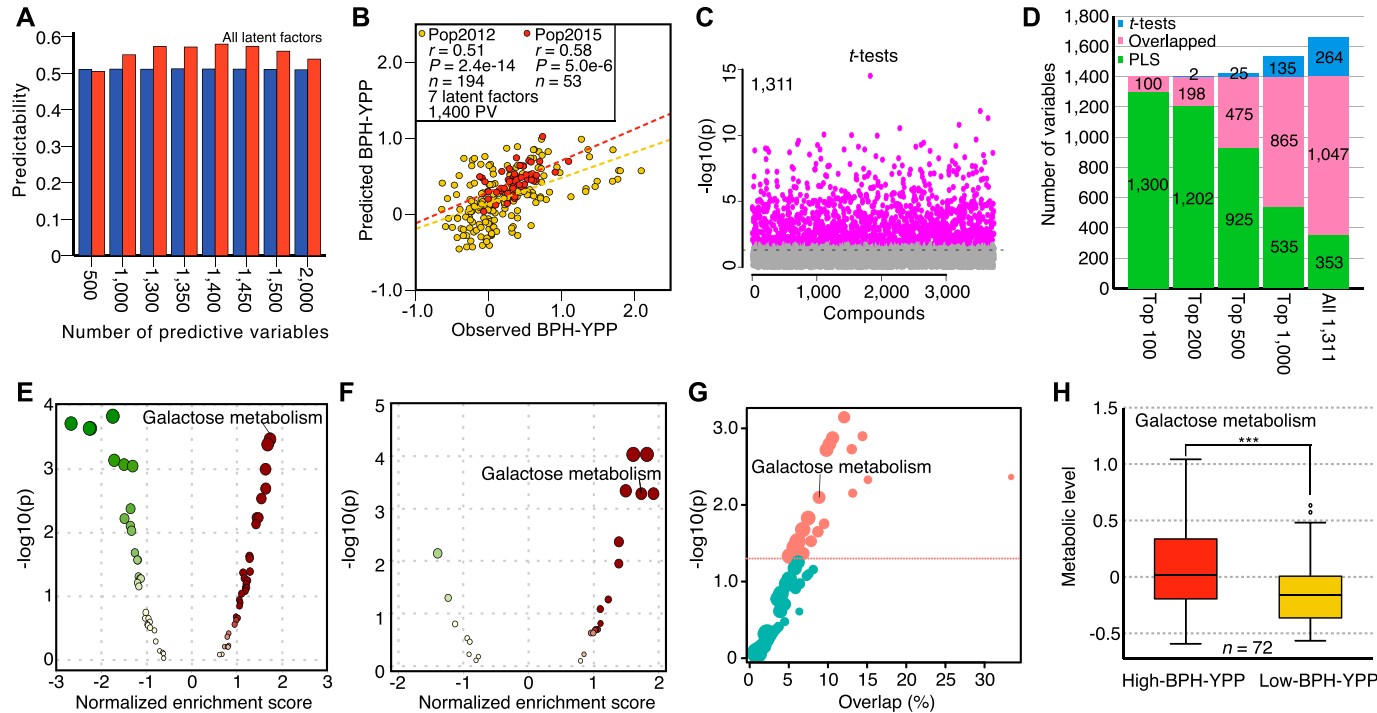

**Figure 4. Feature selection for BPH-YPP.**
**(A)** Predictabilities of BPH-YPP with different numbers of predictive variables selected according to mean VIP values of all seven latent factors. **(B)** Prediction of BPH-YPP for Pop2012 and Pop2015 with the top 1,400 predictive variables selected according to the mean VIP values of the seven latent factors. **(C)** Differential analytes between 72 high- and 72 low-BPH-YPP hybrids from Pop2012. Independent-samples $t$ test, equal group variance assumed, adjusted $P$-value cutoff: 0.05. **(D)** Comparison of the 1,400 predictive variables used in Fig 4B (PLS) and differential analytes identified in Fig 4C ($t$ tests). **(E)** Differential metabolic pathways between high- and low-BPH-YPP hybrids from Pop2012. **(F)** Differential metabolic pathways between high- and low-BPH-YPP hybrids from Pop2015. **(G)** Enriched metabolic pathways between high- and low-BPH-YPP hybrids from Pop2012 based on dysregulated metabolic pathway analysis with MetDNA. **(H)** Comparison of the average levels of metabolites involved in galactose metabolism between high- and low-BPH-YPP hybrids from Pop2012. ***$P$ = 2.7 × $10^{-5}$, two-tailed, independent-samples $t$ test.

These results show that the 1,400 analytes identified in this study are predictive of BPH-YPP and that the up-regulation of galactose metabolism is closely associated with high degrees of yield heterosis.

Next, we substituted the metabolome of Yuetai A (the female parent of Pop2015) with that of Yuetai B (the maintainer line) in the calculation of predictive variables for Pop2015 to increase predictability. However, the predictabilities of BPH-YPP for Pop2015 decreased after switching to the maintainer lines (Fig S11A). Moreover, regarding the effects of changes in training set population size on predictability (Windhausen et al, 2012; Riedelsheimer et al, 2013), half (1/2N) and a quarter (1/4N) of the hybrids from Pop2012 were combined with half (1/2N) of the hybrids from Pop2015 to check for predictability variations. As shown in Fig S11B, the predictabilities of BPH-YPP for Pop2015 decreased in both the 1/2N + 1/2N and 1/4N + 1/2N sets. Thus, the 1/3N + 1/2N training set was the best combination of population sizes for the prediction of yield heterosis in this study.

To further check the stability of the predictive model across conditions, 41 hybrids that were mainly reciprocal and had significant differences between reciprocals in 2012 were phenotyped again in 2015 (Table S19). Because of environmental changes, the BPH-YPPs of the 41 hybrids exhibited significant differences between the 2 yr (Fig 5A and Table S19). Based on the predictive model established in Fig 4B and Table S11, parameters of the 1,400 predictive variables and the constant were used to predict BPH-

YPP for the 41 hybrids. Surprisingly, the correlation coefficient between the observed and predicted values of BPH-YPP was as high as 0.62 (Fig 5B; $P$ = 1.4 × $10^{-5}$), which further confirmed the accurate prediction of yield heterosis for hybrids under different growth conditions.

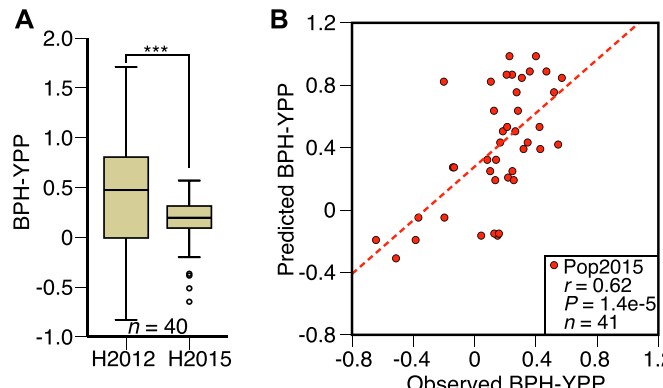

**Figure 5. Validation of the predictive model.**
**(A)** BPH-YPP of reciprocal hybrids phenotyped in 2012 and 2015. ***$P$ = 4.5 × $10^{-4}$, two-tailed, paired-samples $t$ test. **(B)** Prediction of BPH-YPP for 41 hybrids planted in 2015 according to the predictive model built in Fig 4B. The dotted line is the fit line.

**Precision Breeding 1.0 for climate-resilient superior hybrid rice**

Finally, we present a schematic diagram of Precision Breeding 1.0 for guiding the breeding of heterotic and climate-resilient rice based on metabolomics. At least two hybrid populations are used to build predictive models. The first population (Pop1) is the main population, and a half-diallel cross design is recommended. The parental inbred lines should be sufficiently representative; widely planted conventional rice accessions and core accessions from the 3K Rice Genomes Project (Wang et al, 2018) are candidate parents. For parents of another hybrid population (Pop2), either homozygous (conventional, localized, and wild cultivars) or heterozygous (outcrossed and recombinant inbred lines) lines can be the sterile and restorer lines, which provides an advantage in using a wide range of germplasms independently of the degree of heterozygosity and availability of a reference genome. Parental seedlings of both populations were cultivated under uniform conditions. On approximately the 15th day, seedlings were harvested for untargeted LC-MS analysis (Fig S12A). Next, untargeted parental metabolite profiles were obtained, and the mean relative abundances of all detected analytes in every pair of parents were assigned as predictive variables to corresponding hybrids (Fig S12B). Population structure was then dissected by PCA of all predictive variables (Fig S12C). Core hybrids selected from both populations according to principal component 1 scores were combined as a training set, and phenotypic data of core hybrids were collected in field trials (Fig S12D). Subsequently, PLS regression between hybrid performance and corresponding predictive variables was applied to build a predictive model. After filtering low-contribution or unrelated variables according to the mean VIP values of all latent factors, the predictive model was established for both populations (Fig S12E). Then, the performance of "noncore" hybrids was predicted based on parameters of predictive variables (Fig S12F). Finally, hybridizations between parents with predicted good-performance hybrids were conducted, and the candidate hybrids were field validated to determine the elite hybrid combinations (Fig S12G).

# Discussion

The prediction of agronomic traits for single-cross hybrids based on parental differences can transform redundant conventional breeding programs into targeted hybridization tasks, which can significantly reduce the amount of breeding work and improve breeding efficiencies. Here, we found that the average values of parental metabolite levels showed close relationships with hybrid metabolite profiles (Fig 1F), and the means of parental metabolic analytes were predictive of yield heterosis in rice (Fig 1G). A total of 1,400 analytes, some of which are involved in galactose metabolism, were used in the predictive model of BPH-YPP (Figs 4B and E–G). Other metabolic pathways with up- or down-regulated metabolic levels were also found to be associated with yield heterosis in the analyses (Tables S13–17 and Fig S9), suggesting that an optimal balance between these metabolic pathways may contribute to the formation of yield heterosis (Dan et al, 2015). The newly comprehensive metabolite profiling analysis of hybrids

from high- and low-BPH-YPP subgroups will provide more direct proof for the selection of predictive analytes.

For precision breeding of hybrid rice, further research is required to optimize the strategy proposed here. To ensure the stability of predictive models, a larger number of parents, representing collections of rice core germplasms, need to be included in the main population (Pop1). More comprehensive untargeted metabolite profiles (larger numbers of detected peaks) which are obtained in positive ionization mode or the combination of negative and positive ionization data may also influence predictabilities. Notably, the ranges of heterosis that are predicted from models are always much smaller than those observed experimentally (Figs 2B–4B, S5, and S8B); we speculate that this may be caused by unknown factors that can lead to phenotypic variances, such as dynamic environments or interactions between genotype and the environment (Riedelsheimer et al, 2012a; Yu et al, 2016; Li et al, 2018). The breeding of elite hybrid rice involves desirable values for multiple polygenic traits, including heading date, plant height, grain quality, disease, and stress resistance, not just yield. Moreover, determining the effects of training and validation set size on predictability requires more investigation (Windhausen et al, 2012; Riedelsheimer et al, 2013; Schulthess et al, 2017). Hence, robust versions of metabolome-based prediction for multiple polygenic traits remain urgently needed along with extensive and in-depth cooperation among global universities and breeding institutes.

Metabolome-based prediction of agronomic traits has benefited from rapid advancements in metabolite profiling technologies (Blazenovic et al, 2019; Shen et al, 2019). Crop breeders can obtain high-throughput metabolic data without considering the accessibility of reference genomes and ploidy in just a few days, independent of the season. Metabolic markers have demonstrated application potential for both qualitative and quantitative traits in maize (de Abreu et al, 2017; Westhues et al, 2017; Schrag et al, 2018), wheat (Zhao et al, 2015), potato (Steinfath et al, 2010; Sprenger et al, 2018), barley (Heuberger et al, 2014), and *Miscanthus* (Maddison et al, 2017). Therefore, the untargeted metabolome-based prediction strategy, which has wide accessibility, may be further used to breed climate-resilient major and orphan crops, perennial trees, and animals (Hickey et al, 2017) and even perform personalized/localized disease diagnosis for humans (Wray et al, 2013).

# Materials and Methods

**Plant materials**

Details of the plant materials have been presented previously (Dan et al, 2019). Briefly, two hybrid populations were used to explore predictive models of heterosis for rice grain yield. For one population in 2012, 18 *indica*–intermediate type–*japonica* cultivars, some of which have been planted on more than six million hectares in China (including Liaoxing 1 and Wuyunjing 8), were used as the parents with a complete diallel cross design. For another population, recombinant inbred lines ($F_5$) from single- or multiple-cross and backcross hybrids were test-crossed with a Honglian-type cytoplasmic male-sterile line (Yuetai A). Plants of the two populations

were grown separately at different planting densities in 2012 and 2015. In addition, 41 reciprocal hybrids and corresponding parents that were chosen from Pop2012 were planted again under the same conditions as the hybrids in 2015. A randomized block design with three replicates was used for each genotype in each environment. Ten seedlings were planted for each replicate, and the middle five plants of each row were harvested to measure the YPP. Phenotypic data of parents and hybrids are provided in Table S1. The relative BPH-YPP was calculated with the following equation: BPH-YPP = $(F_1 - P_H)/P_H$, where $P_H$ and $F_1$ are the YPP of the high-value parent and hybrid, respectively.

Young rice seedlings used for LC-MS/MS analysis were prepared by a previously reported method (Dan et al, 2019). More than 30 seeds of three pairs of reciprocal hybrids, 18 cultivars, 107 recombinant inbred lines, and Yuetai A were submerged in water at 30°C for 48 h and then transferred to an incubator for germination at 30°C for 24 h. Ten seedlings at the same developmental stage were transplanted into plastic pots with a spacing of 2 × 2 cm. Two biological replicates were used for each genotype. Seedlings were grown at 30°C in a light incubator set to 70% humidity with a 16-h light/8-h dark photoperiod. On the 15th day, ~10 seedlings without parts close to soil (~0.5 cm) and roots were harvested and placed in liquid nitrogen after being washed with ddH$_2$O three times. Two biological replicates of the three pairs of reciprocal hybrids and 18 parents were treated as two independent experimental samples, and two replicates of the 107 recombinant inbred lines, and Yuetai A were combined as a single experimental sample for each genotype. All tissues of each experimental sample were ground into homogenized powders with liquid nitrogen, and 80 mg of powder was transferred into 2-ml EP tubes. After 1 ml of precooled extraction liquid (methanol/acetonitrile/water, 2/2/1, vol/vol/vol) was added with the Agilent Bravo automated liquid handling system (Agilent Technologies), the samples were vortex mixed for 60 s. Ten microliters of all samples were pooled as a quality control sample. Two 30-min ultrasonic treatments were applied, and the tubes were kept at –20°C for 60 min. Then, the samples were centrifuged for 15 min at 14,000$g$ at 4°C. The supernatants were dried in a vacuum concentrator, and 100 $\mu$l of acetonitrile (acetonitrile/water, 1/1, vol/vol) was added to each tube. After vortex mixing, the samples were centrifuged for 15 min at 14,000$g$ at 4°C, and the supernatants were used for subsequent LC-MS analysis.

### UHPLC-Q-TOF MS/MS for untargeted metabolomics analysis

Samples were separated on a Waters (Milford) reversed-phase T3 column (ACQUITY UPLC HSS T3, 2.1 × 100 mm column containing 1.8-$\mu$m particles) using the Agilent 1290 Infinity LC system (Agilent Technologies). After the injection of 2 $\mu$l of the sample, the T3 column was washed with solvent A (0.5 mM ammonium fluoride; Sigma-Aldrich) at a flow rate of 0.3 ml/min at 25°C. The LC gradient elution began with 1% solvent B (acetonitrile; Merck) for 1 min, linearly increased to 100% B in 7 min, was maintained for 2 min at 100% B, was reduced to 1% B in 0.1 min, and was maintained for another 1.9 min at 1% solvent B for re-equilibration. Experimental samples were analyzed in a random order at 4°C, and a quality control sample was injected every nine experimental samples to check the stability of the LC-MS system.

MS data of all samples were collected with a quadrupole time-of-flight mass spectrometer (Agilent 6550 iFunnel QTOF; Agilent Technologies) in negative electron spray ionization mode. The temperature and flow rate of the drying gas were 250°C and 16 L/min, respectively. The temperature and flow rate of the sheath gas were 400°C and 12 L/min, respectively. The pressure of the nebulizer gas was 20 pounds per square inch (psi). The capillary voltage, nozzle voltage, and fragment voltage were 3,000, 0, and 175 V, respectively. The TOF mass spectra data were collected from 60 to 1,200 kD. The acquisition rate was 4 spectra/s and the cycle time was 250 ms.

Subsequently, another quadrupole time-of-flight mass spectrometer (TripleTOF 6600, AB Sciex) was used to acquire tandem mass spectrometry (MS/MS) data of the quality control sample. The parameters of ion source gas1, ion source gas2, and the curtain gas were 40, 80, and 30 psi, respectively. The source temperature was 650°C, and the floating ion spray voltage was –4,500 V. The mass range for the TOF MS scan was 60–1,200 kD, and the quality control sample was injected twice to increase the data acquisition rate with the mass range divided into four sequential windows: 60–200, 190–400, 390–600, and 590–1,200 kD and 200–600, 590–750, 740–900, and 890–1,200 kD. The MS accumulation time was 200 ms/spectra and the tandem mass spectra were collected in an information-dependent manner in high-sensitivity mode. The declustering potential was –60 V, and the collision energy was 35 ± 15 eV. Ten candidate ions were monitored per cycle using an isotope isolation width of 4.0 kD, and the MS/MS accumulation time was 50 ms.

### Quantification of metabolomics data

The msconvert function in ProteoWizard (Chambers et al, 2012) was used to convert raw MS files (.d) to the mzXML format. Then, these files were uploaded to the XCMS (Gowda et al, 2014) for data processing, including feature detection, retention time correction, and alignment with the method described previously (Jia et al, 2018). The MS/MS data were matched against the in-house standard spectral library (Wang et al, 2016) and lipid MS/MS spectral library (Tu et al, 2017), and metabolites were identified with accurate mass (<25 ppm). To align the MS and MS/MS data, m/z errors less than 15 ppm and retention time errors fewer than 20 s were applied. To check the reliability of the metabolomics data, a PCA was performed on both the experimental and quality control samples before the data normalization procedure. The location of the quality control sample in the three-dimensional histogram and plots of the top five principal components indicated that the obtained metabolite profiles were accurate and appropriate for subsequent analyses (Fig S13). Normalization of the relative abundances of the 3,746 detected analytes was performed with MetaboAnalyst 4.0 (Xia & Wishart, 2011). Samples with two biological replicates were averaged first. Sample normalization (normalization by sum), data transformation (none), and data scaling (autoscaling) were performed with MetaboAnalyst 4.0 to make the samples and features suitable for statistical analyses. We transformed parental metabolite levels to obtain the means, differences, and ratios of parental metabolite files for each hybrid with equations $F_{Means} = (P_1 + P_2)/2$, $H_{Differences} = P_1 - P_2$, and $F_{Ratios} = P_1/P_2$, where F, $P_1$, and $P_2$ are the metabolic levels of the hybrid, female

parent, and male parent, respectively. Some of the metabolic data were reported in a previous study (Dan et al, 2019), and the complete metabolic data corresponding to all parents and hybrids used for analyses are provided in Table S3, which was prepared according to recommendations (Fernie et al, 2011).

### Determination of population structure and selection of predictive analytes

The 3,746 detected analytes and 34 pairs of InDel markers were used to assess the genetic differences of parents of Pop2012. Dendrograms were drawn based on the relative abundances of detected analytes and genotypes of each marker with the Statistical Analysis module (with Euclidean and Ward as the distance measure and clustering algorithm, respectively) on MetaboAnalyst. PCA was performed on the metabolome of parents of both populations. The mean relative abundances of the 3,746 detected analytes in every pair of parents were calculated as predictive variables for corresponding hybrids. PCA was performed to demonstrate the population distances. Scores for the first principal component were reordered from high to low, and hybrids were chosen every two, three, or four intervals as the core hybrids.

In the exploration of predictive models for BPH-YPP, hybrids from Pop2012 were divided into two subgroups based on low- and high-BPH-YPP values. PCA and PLS-DA were performed on hybrids in the low- and high-BPH-YPP subgroups to investigate their metabolic relationships. Analytes were screened according to VIP values from the results of PLS-DA, and analytes with low VIP values were removed from the models. Subsequently, core hybrids from both populations were combined as a training set, and PLS was applied to bridge the predictive variables and hybrid performance. The BPH-YPP of the hybrids and the predictive variables were dependent and independent variables, respectively. The number of latent factors was selected when the largest adjusted R-square value of cumulative Y variance emerged. The mean VIP values of all latent factors were used to screen predictive analytes. Variables with large contributions in the model were retained for modeling, and the number of predictive analytes was adjusted to achieve high predictability.

### Pathway mapping of metabolic analytes

Differential analytes with m/z, P-values, and t scores from t tests comparing high- and low-BPH-YPP hybrids were mapped to metabolic pathways through the module "MS Peaks to Pathways" on MetaboAnalyst (Xia & Wishart, 2011). The mass accuracy was 5.0 ppm. The analytical mode was negative, and the data format was three-column according to the data preparation instructions. GSEA was the algorithm, and the *Oryza sativa japonica* (Japanese rice) pathway library was selected for enrichment analysis. Dysregulated network analysis was performed with MetDNA (Shen et al, 2019). The tandem mass spectra data with a mass range of 60–1,200 kD collected from the quality control sample were uploaded for metabolite identification. Hybrids with high- and low BPH-YPPs were grouped into the control and case groups, respectively. The ionization polarity was negative, and reversed-phase LC was used. The mass spectrometer instrument type was Sciex TripleTOF, and the collision energy was 35 ± 15 eV. t test was selected as the statistical method. The library of *Arabidopsis thaliana* (thale cress) was chosen for enrichment analysis. The cutoff P-value was 0.05, and the P-values were corrected with a false discovery rate in the analysis.

### Statistical analysis

Independent-samples t test (two-tailed) was performed to compare differences of YPP for parents and hybrids, the BPH-YPP for hybrids from both populations, and the average levels of metabolites in pathways between high- and low-BPH-YPP hybrids with SPSS (IBM SPSS Statistics for Windows, Version 20.0; IBM Corp.). Levene's test was used to determine the equality of variances. Paired-samples t test (two-tailed) was used to compare differences of BPH-YPP between reciprocal hybrids that were selected from population 2012 and hybrids that were phenotyped in both years. PLS and bivariate correlation analyses were implemented with SPSS. PCA and PLS-DA were conducted with the "Statistical Analysis" module on MetaboAnalyst (Xia & Wishart, 2011). The correlation coefficient of Pearson correlation (two-tailed) between the observed and predicted BPH-YPP values was treated as predictability.

### Data access

Raw metabolomics files generated in this study have been deposited to the metabolomics database MetaboLights (www.ebi.ac.uk/metabolights) under the study identifier MTBLS742.

## Supplementary Information

## Acknowledgements

We thank members of the 3134 laboratory for the assistance of field experiments and valuable discussions. This research was supported by the National Key Research & Development Program of China (Grant No 2017YFD0100400), National Natural Science Foundation of China (Grants 31771746 and 31801439), National Rice Industry Technology System (Grant No CARS-01-07), and the China Postdoctoral Science Foundation.

### Author Contributions

Z Dan: conceptualization, resources, software, funding acquisition, investigation, methodology, and writing—original draft, review, and editing.
Y Chen: data curation, software, investigation, and methodology.
W Zhao: data curation and investigation.
Q Wang: data curation and investigation.
W Huang: conceptualization, resources, data curation, supervision, funding acquisition, investigation, and writing—original draft, review, and editing.

## Conflict of Interest Statement

The authors declare that they have no conflict of interest.

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
