## [Reviewer comments · Life Science Alliance]

Life Science Alliance

Metabolome-based prediction of yield heterosis contributes to the breeding of elite rice

Zhiwu Dan, Yunping Chen, Weibo Zhao, Qiong Wang, and Wenchao Huang

DOI: <https://doi.org/10.26508/lsa.201900551>

Corresponding author(s): Zhiwu Dan, Wuhan University

Review Timeline:

Submission Date:	2019-09-15
Editorial Decision:	2019-10-02
Revision Received:	2019-11-05
Editorial Decision:	2019-11-18
Revision Received:	2019-11-24
Accepted:	2019-11-26

Scientific Editor: Andrea Leibfried

Transaction Report:

October 2, 2019

Re: Life Science Alliance manuscript #LSA-2019-00551-T

Zhiwu Dan
Wuhan University

Dear Dr. Dan,

Thank you for submitting your manuscript entitled "Metabolome-based prediction of yield heterosis contributes to the breeding of heterotic and adaptive rice" to Life Science Alliance. The manuscript was assessed by expert reviewers, whose comments are appended to this letter.

As you will see, the reviewers appreciate your approach but think that the work is not presented in a way that will allow others to access and make use of it. We would thus like to invite you to submit a revised version of your manuscript, addressing the individual points raised by the reviewers and, importantly, this major concern. The reviewers provide constructive input on how to do so, but please get in touch in case you would like to discuss revision points further.

Thank you for this interesting contribution to Life Science Alliance. We are looking forward to receiving your revised manuscript.

Sincerely,

Andrea Leibfried, PhD
Executive Editor
Life Science Alliance
Meyerhofstr. 1
69117 Heidelberg, Germany

t +49 6221 8891 502
e a.leibfried@life-science-alliance.org
www.life-science-alliance.org

B. MANUSCRIPT ORGANIZATION AND FORMATTING:

Reviewer #1 (Comments to the Authors (Required)):

The authors present a very interesting study aiming to utilize non-targeted metabolite profiles to predict yield heterosis. If successful, this type of approach could be an extremely valuable tool for breeders. While the concept of the paper is of high interest and value, the results are presented in such a confusing way it is very difficult for the reader to understand what was done or to effectively interpret the results. Specific comments are below; however, I think that the paper needs an overall reorganization to enable effective communication of the results. Furthermore, for improved readability by both the breeding and metabolomics audiences some greater explanation of specific terms and applications in these two different fields may be required. In general, unless presented as a methods study, only the final and best modeling conditions should

be included in the manuscript. This value of this manuscript would be greatly increased by simplification and a focus on presentation of only the results that tell the final story.

- How did the authors calculate "means, differences, and ratios" of metabolite levels? These should be clearly described as they are not standard methods of reporting metabolite data. Further, it is unclear what the value of these different measures is in the interpretation. The interpretation/value of Figure 1a and b is unclear.

- I do not understand Figure 1G.

- Figures 2 and 3 contain too much information. I don't think that showing the PCA component matrix is needed.

- How were the "core" and "noncore" hybrids determined?

- I do not understand the conclusion "the first step for the prediction of yield heterosis is the dissection of population structures". What does this mean?

- Why was the metabolite analysis performed using negative ionization? Typically, positive ionization will result in much richer metabolite profiles of plant tissue. Further, this choice will bias the types of molecules that are detected. This choice should be discussed.

- This statement is not informative: "The distribution of quality control sample in the three-dimensional histogram and the top five principal components plots indicated that the obtained metabolite profiles were accurate and appropriate for subsequent analyses." What does this mean? Without seeing this data how will the reader evaluate that the profiles were "accurate and appropriate"?

- The method used for transformation and normalization needs to be described.

- How were the datasets aligned so that MS/MS data from the Triple TOF could be related to specific compounds detected in profiling data from the TOF?

- Was a multiple testing correction applied for the t-test results, not just in pathway analysis but also for other metabolite analyses? Also, clarification of which statistical comparisons were made is needed. Consider using ANOVA or multivariate ANOVA where appropriate.

Reviewer #2 (Comments to the Authors (Required)):

This paper describes an approach to predict heterosis effects on grain yield in rice, following up on an earlier paper from the same group, where grain size was the target phenotype. Of course, yield would in most cases be the much more interesting trait for any breeder, but is also genetically much more complex. The authors show that plant yield of heterotic offspring can be reasonably well predicted from metabolite profiles of the parents, using young seedlings grown under controlled conditions. There are, however, several points the authors need to consider:

1) What is an "adaptive crop"? A crop adapted to a particular environment, or one that can adapt to different conditions? This should be rephrased for greater clarity.

2) What are "genome-scale metabolite profiles"? I understand genome-scale transcript profiles, but I am not sure of the meaning of this term. Certainly, current metabolomics covers only a small fraction of the metabolites of any organism, i.e. of those which are, albeit indirectly, encoded in its genome.

3) p. 2: the sentence starting with "The metabolite levels..." should be rephrased. It is unclear to me what it says.

4) p. 5: "...traits such as yield and disease..." - should this be disease resistance? Disease is not a trait.

5) p. 7: "...metabolite profiles can roughly reflect differences at the genomic level." While there are some similarities, there are also strong differences. This should be described more realistically.

6) Fig. 4H and L: the y-axes need labels.

7) p. 21: how many metabolites are included in "galactose metabolism"? "Levels of galactose metabolism" - this should probably be levels of metabolites in galactose metabolism. How many of these metabolites (what fraction) were significantly different between the two groups? I find it a bit doubtful to include 3PGA, Glc, Fru and Suc in Gal metabolism, as these are compounds that mainly play a dominant role in central carbohydrate metabolism.

8) p. 25: "schematic diagram" - I was not able to find that in the manuscript.

9) p. 27: "pathways with different expression levels" - what is meant here? Pathways do not have expression levels (they are not genes).

10) p. 27: "We hypothesize..." - do the present data provide any evidence for this hypothesis?

11) In all cases, the range of heterosis that is predicted from the models is much smaller than that experimentally observed (by about a factor of 2). This fact needs to be critically discussed. Why is there such a systematic underestimation of heterosis in the models?

12) Does the experimentally observed range of heterosis have an influence on predictability? The 2015 population was smaller and had a lower range of heterosis and poorer predictability. What happens to the prediction quality of the 2012 population if predictions are only made for a population consisting of lines with BPH-YPP below 1?

Point-by-point response to comments

Reviewer comments:

Reviewer #1

The authors present a very interesting study aiming to utilize non-targeted metabolite profiles to predict yield heterosis. If successful, this type of approach could be an extremely valuable tool for breeders. While the concept of the paper is of high interest and value, the results are presented in such a confusing way it is very difficult for the reader to understand what was done or to effectively interpret the results. Specific comments are below; however, I think that the paper needs an overall reorganization to enable effective communication of the results. Furthermore, for improved readability by both the breeding and metabolomics audiences some greater explanation of specific terms and applications in these two different fields may be required. In general, unless presented as a methods study, only the final and best modeling conditions should be included in the manuscript. This value of this manuscript would be greatly increased by simplification and a focus on presentation of only the results that tell the final story.

Answer: We greatly appreciate the reviewer for a careful and thorough reading of our manuscript and valuable suggestions, which have helped us to improve the quality of our manuscript. As suggested by the reviewer, we have merged and simplified the result section for clearer presentation (**p. 6, lines 81-85, 90-99; p. 7, lines 102-106; p. 8, lines 114-115, 121, 126-133; p. 9, lines 141-145, p. 10, lines 151-153, 156-159; p.**

11, lines 167-171; p. 12, lines 179-182; p. 13, lines 206-208; p. 15, lines 225-230, 233-237); added contents to corresponding required parts (p. 5, lines 72-74; p. 12, lines 192-193; p. 14, lines 213-217; p. 25, lines 400-403; p. 26, lines 414-418; p. 27, lines 420-423, 425-431; p. 28, lines 446-447); reorganized all the main figures to enable effective communication of the results, and moved some figures that generated in the analyzing procedures to supplemental files to highlight final findings (**Fig S1, Fig S2, Fig S5-8, and Fig S11**). Meanwhile, we have changed the manuscript title to “**Metabolome-based prediction of yield heterosis contributes to the breeding of elite rice**” to make it more appropriate according to the Author guideline of Life Science Alliance (The total length of the title should not exceed 100 characters). Please find below a detailed response to the points raised.

1) How did the authors calculate "means, differences, and ratios" of metabolite levels? These should be clearly described as they are not standard methods of reporting metabolite data. Further, it is unclear what the value of these different measures is in the interpretation. The interpretation/value of Figure 1a and b is unclear.

Answer: We have followed the reviewer’s suggestion and described details about how to calculate the means, differences, and ratios of parental metabolite levels in the section of *Materials and Methods*. We have included the following text: “*We transformed parental metabolite levels to get the means, differences, and ratios of*

parental metabolite files for each hybrid with the equations: $F_{Means} = (P_1 + P_2)/2$, $H_{Differences} = P_1 - P_2$, $F_{Ratios} = P_1/P_2$, where F , P_1 , and P_2 are the metabolic levels of the hybrid, female parent and male parent, respectively” (p. 27, lines 428-431).

To find an appropriate manner for transforming parental metabolite levels as predictive variables for each hybrid from the two hybrid populations, we calculated the means, differences, and ratios of parental metabolite levels and analyzed the relationship between parental metabolite levels and hybrid metabolite profiles (see Figure 1E and 1F which were Figure 1A and 1B in the original version). Figure 1E and 1F showed that the means and differences of parental metabolite levels manifested tighter connections to hybrid metabolite profiles, compared to the ratios of parental metabolite levels. Next, we performed a partial least squares regression analysis between the means and differences of parental metabolite levels and better-parent heterosis for yield per plant. In the regression procedures, changed numbers of latent factors generated different R-square values which can be understood as the proportions of yield heterosis’ variance explained by the transformed parental metabolite levels. We found that the means of parental metabolite levels were more suitable to be predictive variables of yield heterosis, compared to the differences (see Figure 1G).

To help the interpretation of Figure 1, we reorganized the order of contents in Figure 1 and revised corresponding descriptions in the first result (p. 6, lines 82-84, 90-99, 102-106).

Figure 1 Determination of predictive variables for BPH-YP.

2) I do not understand Figure 1G.

Answer: Please see our response above.

3) Figures 2 and 3 contain too much information. I don't think that showing the PCA component matrix is needed.

Answer: We have followed the reviewer's suggestion and have removed score plots in Figure 2 and Figure 3 to make the results easier to interpret. Please see Figure 2 and Figure 3 in the revised manuscript.

4) *How were the "core" and "noncore" hybrids determined?*

Answer: The “core” and “noncore” hybrids from each population were chosen according to PCA scores in Table S8 and Table S9. We performed PCA on 3,746 predictive variables, and scores of the first principal component (PC 1) were reordered from high to low. Core hybrids were chosen every two, three or four intervals, and the rest were treated as noncore hybrids. We have described the details in the section of *Materials and Methods* (p. 28, lines 446-447).

5) *I do not understand the conclusion "the first step for the prediction of yield heterosis is the dissection of population structures". What does this mean?*

Answer: According to our results, we found that the dissection of population structures was the initial critical step in multivariate modeling for the prediction of yield heterosis. To avoid ambiguity, we have revised the sentence to “*the dissection of population structures as the initial step for building predictive models is better than the screening of predictive variables*” (p. 11, lines 175-177).

6) *Why was the metabolite analysis performed using negative ionization? Typically, positive ionization will result in much richer metabolite profiles of plant tissue. Further, this choice will bias the types of molecules that are detected. This choice should be discussed.*

Answer: We used the untargeted metabolite profiles for building predictive models of yield heterosis in this study. We first dissected the population structures based on the full-scale metabolite profiles and selected core hybrids for multivariate modeling. Then we filtered low-contribution and unrelated analytes to increase predictabilities. Finally, we achieved robust predictions for hybrids under different growth conditions, and the predictabilities were close to 0.6. As commented by the reviewer, metabolite profiles in positive ionization mode may influence the predictability. To make the effects of metabolite profiles on predictability more comprehensive, we have included the following text in the *Discussion* section: *“More comprehensive untargeted metabolite profiles (greater numbers of detected peaks) which are obtained in positive ionization mode or the combination of both negative and positive ionization data may also influence predictabilities”* (p. 19, lines 300-303).

7) *This statement is not informative: "The distribution of quality control sample in the three-dimensional histogram and the top five principal components plots indicated that the obtained metabolite profiles were accurate and appropriate for*

subsequent analyses." What does this mean? Without seeing this data how will the reader evaluate that the profiles were "accurate and appropriate"?

Answer: To prove that the obtained raw metabolite files were accurate and appropriate for analyses, we performed PCA on metabolic data of the quality control and experimental samples. We have added PCA score plots of quality control sample in Figure S13 which demonstrated that the metabolic data were accurate and reliable.

Figure S13. PCA score plots of quality control and experimental samples.

8) The method used for transformation and normalization needs to be described.

Answer: As suggested, we have included the following text in the section of *Materials and Methods*: “Sample normalization (normalization by sum), data transformation (none) and data scaling (auto scaling) were performed with the

MetaboAnalyst 4.0 to make the samples and features suitable for statistical analyses”

(p. 27, lines 425-428).

9) *How were the datasets aligned so that MS/MS data from the Triple TOF could be related to specific compounds detected in profiling data from the TOF?*

Answer: We have followed the reviewer’s suggestion and have provided this information in the part of *Materials and Methods* and added two references for interpretation. About the details of the alignment between MS/MS data in experimental samples and the spectral libraries, we have included the following text in our manuscript: *“The MS/MS data were matched against the in-house standard spectral library (Wang et al. 2016) and lipid MS/MS spectral library (Tu et al. 2017), and metabolites were identified with accuracy mass (<25 ppm). To align the MS and MS/MS data, m/z errors less than 15 ppm and the retention time errors less than 20 seconds were applied”* **(p. 26, lines 414-418).**

10) *Was a multiple testing correction applied for the t-test results, not just in pathway analysis but also for other metabolite analyses? Also, clarification of which statistical comparisons were made is needed. Consider using ANOVA or multivariate ANOVA where appropriate.*

Answer: In this study, the statistical comparisons performed in other metabolomics analyses were mainly two-group comparisons (high- and low-BPH-YPP two groups). So we used independent samples *t*-test, and the types of statistical comparisons have been indicated in the results or figure legends.

We thank the Reviewer for all the constructive comments on effective communication of the results and improving the readability of this manuscript.

Reviewer: 2

This paper describes an approach to predict heterosis effects on grain yield in rice, following up on an earlier paper from the same group, where grain size was the target phenotype. Of course, yield would in most cases be the much more interesting trait for any breeder, but is also genetically much more complex. The authors show that plant yield of heterotic offspring can be reasonably well predicted from metabolite profiles of the parents, using young seedlings grown under controlled conditions. There are, however, several points the authors need to consider:

1) What is an "adaptive crop"? A crop adapted to a particular environment, or one that can adapt to different conditions? This should be rephrased for greater clarity.

Answer: We have followed the reviewer's suggestion and have changed the "adaptive crops" to "crops adaptive to different conditions" to make the meaning clear in the sections of *Abstract* and *Introduction* (**p. 2, lines 21-23; p. 3, lines 44-46**).

Meanwhile, we have changed the title to “**Metabolome-based prediction of yield heterosis contributes to the breeding of elite rice**” to make it more appropriate and meet the journal’s demand (The total length of the title should not exceed 100 characters).

2) *What are "genome-scale metabolite profiles"? I understand genome-scale transcript profiles, but I am not sure of the meaning of this term. Certainly, current metabolomics covers only a small fraction of the metabolites of any organism, i.e. of those which are, albeit indirectly, encoded in its genome.*

Answer: We agree with the reviewer and have corrected the “genome-scale” to “untargeted” to make the meaning more precise in the *Abstract* section (**p. 2, lines 26-28**).

3) *p. 2: the sentence starting with "The metabolite levels..." should be rephrased. It is unclear to me what it says.*

Answer: We agree with the reviewer and have rephrased the sentence to make it interpretable in the *Abstract*: “*Metabolites belonging to specific pathways were tightly connected to yield heterosis, and the up-regulation of galactose metabolism represent robust yield heterosis for hybrids across different growth conditions*” (**p. 2, lines 30-32**).

4) p. 5: "...traits such as yield and disease..." - should this be disease resistance?

Disease is not a trait.

Answer: We have followed the reviewer's suggestion and have added the word "resistance" to this sentence (p. 5, lines 72-74).

5) p. 7: "...metabolite profiles can roughly reflect differences at the genomic level."

While there are some similarities, there are also strong differences. This should be described more realistically.

Answer: We agree with the reviewer and have revised this sentence to avoid misunderstanding through removing the attributive clause: "A dendrogram of parents from Pop2012 based on parental metabolite profiles showed a similar clustering trend to that based on indica-japonica-specific insertion and deletion DNA markers (Fig 1C and Fig S1)" (p. 6, lines 90-93).

6) Fig. 4H and L: the y-axes need labels.

Answer: We have followed the reviewer's suggestion and the y-axes labels have been added to Figures 4H and L which were Figures 4D and H in the revised manuscript.

Figure 4. Feature selection for BPH-YPP.

7) p. 21: how many metabolites are included in "galactose metabolism"? "Levels of galactose metabolism" - this should probably be levels of metabolites in galactose metabolism. How many of these metabolites (what fraction) were significantly different between the two groups? I find it a bit doubtful to include 3PGA, Glc, Fru and Suc in Gal metabolism, as these are compounds that mainly play a dominant role in central carbohydrate metabolism.

Answer: As shown in the result of metabolite set enrichment analysis on difference analytes (using the module "MS peaks to pathways" on MetaboAnalyst; <https://www.metaboanalyst.ca>) from Pop2012 (Table S13), 11 metabolites were included in galactose metabolism. We agree with the reviewer and have revised the

“*levels of galactose metabolism*” to “*levels of metabolites in galactose metabolism*” in the manuscript to make the description more accurate (**p. 14, lines 213-215**).

The number of metabolites included in galactose metabolism with quantitative information was four, and all these metabolites (glycerol, D-Sorbitol, 2-Dehydro-3-deoxy-D-galactonate, and 3-beta-D-galactosyl-sn-glycerol) had significantly different metabolic levels between the two groups (Table S18) (**p. 14, lines 215-217**).

The carbohydrate metabolism contains glycolysis/gluconeogenesis, citrate cycle (TCA cycle), fructose and mannose metabolism, **galactose metabolism**, starch and sucrose metabolism, glyoxylate and dicarboxylate metabolism, and so on. We found the metabolites (3PGA, Glc, Fru and Suc) were included in the galactose metabolism (osa00052, Galactose - *Oryza sativa japonica*; https://www.genome.jp/dbget-bin/www_bget?pathway:osa00052). Meanwhile, we agree with the reviewer that these metabolites may be involved in other metabolic pathways with different functions.

8) p. 25: "*schematic diagram*" - *I was not able to find that in the manuscript.*

Answer: Based on the findings in this study, we put forward a schematic diagram that was shown in Figure S12 in the supplemental files.

Figure S12. Schematic diagram of Precision Breeding 1.0.

9) p. 27: "pathways with different expression levels" - what is meant here? Pathways do not have expression levels (they are not genes).

Answer: We have followed the reviewer's suggestion and have corrected the "expression levels" to "metabolic levels" (p. 19, lines 290-292).

10) p. 27: "We hypothesize..." - do the present data provide any evidence for this hypothesis?

Answer: According to the result of metabolite set enrichment analysis of difference analytes from Pop2012, a total of 11 metabolic pathways were enriched with positive or negative normalized enrichment scores (Table S13). Meanwhile, metabolic levels

of all the six metabolic pathways were up- or down-regulated, correspondingly (Table S15, Table S16, Table S17, Figures 4E-H, and Figure S9). Hence, we doubt that there may exist an optimal balance between these metabolic pathways in the formation of yield heterosis. We have included the following text in the section of *Discussion* to make meaning smooth: “Other metabolic pathways with up- or down-regulated metabolic levels are also found to be associated with yield heterosis in the analyses (Table S13, Table S14, Table S15, Table S16, Table S17, and Fig S9), suggesting that an optimal balance between these metabolic pathways may contribute to the formation of yield heterosis” (p. 19, lines 290-293).

Figure S9. Metabolite set enrichment analysis of difference analytes from Pop2012.

11) *In all cases, the range of heterosis that is predicted from the models is much smaller than that experimentally observed (by about a factor of 2). This fact needs to be critically discussed. Why is there such a systematic underestimation of heterosis in the models?*

Answer: As indicated by the reviewer, the ranges of heterosis that are predicted from models are always much smaller than that experimentally observed (Figure 2, Figure 3, Figure 4B, Figure S5, and Figure 8B). This phenomenon also has been reported in some other studies with different predictive methods (Riedelsheimer et al. 2012; Yu et al. 2016; Li et al. 2018). We have included the following text in the *Discussion* section for this interesting point: “*Notably, the ranges of heterosis that are predicted from models are always much smaller than that experimentally observed (Fig 2, Fig 3, Fig 4B, Fig S5, and Fig 8B), we speculate that this may be caused by unknown factors that can lead phenotypic variances from the dynamic environments or interactions between genotype and environment (Riedelsheimer et al. 2012a; Yu et al. 2016; Li et al. 2018)*” (p. 19, lines 303-307).

12) *Does the experimentally observed range of heterosis have an influence on predictability? The 2015 population was smaller and had a lower range of heterosis and poorer predictability. What happens to the prediction quality of the 2012 population if predictions are only made for a population consisting of lines with BPH-YPP below 1?*

Answer: Although the range of heterosis for Pop2015 is narrower than that of Pop2012 (Figure 1B), the predictability of yield heterosis for Pop2015 ($r = 0.58$) was not lower than that of Pop2012 ($r = 0.51$) after adopting appropriate predictive variable screening method (Figure 4B). Meanwhile, we have followed the reviewer's suggestion and have analyzed BPH-YPP for Pop2015. While only one hybrid from Pop2015 was with BPH-YPP larger than 1, suggesting that the predictability will not be influenced by the range of heterosis. We have included the following text in the *Results* section to show this point: “*Thus, hybrids from both populations were predicted with predictabilities close to 0.6 though the ranges of yield heterosis for the two populations differed largely*” (p. 12, lines 192-193).

We greatly appreciate the reviewer for rightful suggestions on details of the predictive strategy proposed in the study, and thanks again for your time reviewing our manuscript.

References

- Li X, Guo T, Mu Q, Li X, Yu J (2018) Genomic and environmental determinants and their interplay underlying phenotypic plasticity. *Proc Natl Acad Sci USA* 115: 6679-6684. doi:10.1073/pnas.1718326115.
- Riedelsheimer C, Czedik-Eysenberg A, Grieder C, Lisec J, Technow F, Sulpice R, Altmann T, Stitt M, Willmitzer L, Melchinger AE (2012) Genomic and metabolic prediction of complex heterotic traits in hybrid maize. *Nat Genet* 44: 217-220. doi:10.1038/ng.1033.
- Yu X, Li X, Guo T, Zhu C, Wu Y, Mitchell SE, Roozeboom KL, Wang D, Wang ML, Pederson GA et al (2016) Genomic prediction contributing to a promising global strategy to turbocharge gene banks. *Nat Plants* 2: 16150. doi:10.1038/nplants.2016.150.

November 18, 2019

RE: Life Science Alliance Manuscript #LSA-2019-00551-TR

Dr. Zhiwu Dan
Wuhan University
No. 299, Bayi road, Wuchang District
Wuhan, Hubei 430072
China

Dear Dr. Dan,

Thank you for submitting your revised manuscript entitled "Metabolome-based prediction of yield heterosis contributes to the breeding of elite rice". As you will see, while some text changes are still required, the reviewers are overall in favor of publication now. We would thus be happy to publish your paper in Life Science Alliance pending final revisions:

- Please address the remaining reviewer comments by further text changes.
- Please make sure that the author order is the same in our submission system and in the manuscript itself.
- Please make sure that all corresponding authors link their profile in our submission system to their ORCID iDs.

A. FINAL FILES:

-- Summary blurb (enter in submission system): A short text summarizing in a single sentence the study (max. 200 characters including spaces). This text is used in conjunction with the titles of

papers, hence should be informative and complementary to the title. It should describe the context and significance of the findings for a general readership; it should be written in the present tense and refer to the work in the third person. Author names should not be mentioned.

B. MANUSCRIPT ORGANIZATION AND FORMATTING:

Sincerely,

Reviewer #1 (Comments to the Authors (Required)):

While the authors have attempted to improve the manuscript, it remains difficult to read and contains too much details that detract from presentation of the final results. For example, much of the details on pages 8-13 could be removed and condensed into a few sentences that describe the final optimized approach.

Reviewer #2 (Comments to the Authors (Required)):

The authors have revised their paper following the recommendations of the reviewers and as a result, the paper is strongly improved. There are still some minor language problems.

Point-by-point response to comments

Reviewer comments:

Reviewer #1 (Comments to the Authors (Required)):

While the authors have attempted to improve the manuscript, it remains difficult to read and contains too much details that detract from presentation of the final results. For example, much of the details on pages 8-13 could be removed and condensed into a few sentences that describe the final optimized approach.

Answer: To make the manuscript concise and easier for understanding, we have simplified descriptions about the exploitation procedures for building predictive models in the second result (**p. 8, lines 125-128; p. 9, lines 129-142; p. 10, 146-153**). Meanwhile, we have shortened the section of “Optimization and validation of the predictive model” and corresponding necessary contents have remained in the manuscript (**p. 11, lines 167-169, 174-176; p. 12, 177-184; p. 13, 197**). In addition, we have addressed language errors for the entire manuscript. The text effects of the presentation of final results in the revised version have been further enhanced as suggested by the reviewer. Thanks again for your patience in improving the clarity and readability of our manuscript.

Reviewer #2 (Comments to the Authors (Required)):

The authors have revised their paper following the recommendations of the reviewers and as a result, the paper is strongly improved. There are still some minor language problems.

Answer: As suggested, we have carefully corrected the spelling errors and used the America Journal Experts English language polishing service to improve the writing of the entire manuscript. Thanks for the suggestion.

November 26, 2019

RE: Life Science Alliance Manuscript #LSA-2019-00551-TRR

Dr. Zhiwu Dan
Wuhan University
No. 299, Bayi road, Wuchang District
Wuhan, Hubei 430072
China

Dear Dr. Dan,

Thank you for submitting your Research Article entitled "Metabolome-based prediction of yield heterosis contributes to the breeding of elite rice". It is a pleasure to let you know that your manuscript is now accepted for publication in Life Science Alliance. Congratulations on this interesting work.

DISTRIBUTION OF MATERIALS:

Again, congratulations on a very nice paper. I hope you found the review process to be constructive and are pleased with how the manuscript was handled editorially. We look forward to future exciting submissions from your lab.

Sincerely,
